# Eccentricity forcing on Tropical Ocean Seasonality

Luc Beaufort[1] and Anta Clarisse Sarr[2,3]

[1]Aix-Marseille Univ., CNRS, IRD, INRAE, CEREGE, 13090 Aix-en-Provence, France
[2]Univ. Grenoble Alpes, Univ. Savoie Mont Blanc, CNRS, IRD, Univ. Gustave Eiffel, ISTerre, 38000 Grenoble, France
[3]now at: Department of Earth Sciences, University of Oregon, Eugene, 97403, USA

**Correspondence:** Luc Beaufort (beaufort@cerege.fr)

**Abstract.** The amount of radiative energy received at the Earth's surface depends on two factors: Earth-Sun distance and sunlight angle. Because of the former, high eccentricity cycles can induce the appearance of seasons in the tropical ocean. In this paper, we use the Earth System model IPSL-CM5A2 to investigate the response of the low-latitude ocean to variations in Earth's orbit eccentricity. Sea Surface Temperature (SST) and Primary Production (PP) were simulated under six precession configurations at high eccentricity and two configurations at low eccentricity, representing extreme configurations observed over the past million years. Results show that high eccentricity leads to increased seasonality in low latitude mean SST, with an annual thermal amplitude of approximately 2.2°C (vs. 0.5°C at low eccentricity). Low latitude mean PP, which already exhibits inherent seasonality under low eccentricity conditions, sees its seasonality largely increased under high eccentricity. As a consequence, we show that on long time scales the intensity of SST seasonality exhibits only the eccentricity frequency, whereas that of PP additionally follows precession dynamics. Furthermore, the seasonal variations in both SST and PP at high eccentricities are influenced by the annual placement of perihelion with its direct impact of radiative energy received in tropical regions. This leads to a gradual and consistent transition of seasons within the calendar. We introduce the concept of "eccentriseasons," referring to distinct annual thermal differences observed in tropical oceans under high eccentricity conditions, which shift gradually throughout the calendar year. These findings have implications for understanding low latitude climate phenomena such as El Niño-Southern Oscillation and monsoons in the past.

## 1   Introduction

Seasonality is one of the central characteristics of climate. It relates to the geometry of the Earth's orbit around the Sun and the rotational axis configuration, and the effect of these on the amount and distribution of radiative energy received at the Earth's surface (Laepple and Lohmann, 2009; Milankovitch, 1941). The seasonality at a given latitude is largely determined by the Earth-Sun distance and the angle at which sunlight reaches the Earth's surface, both defining the amount of radiative energy received seasonally and locally. Both parameters are influenced by variations in obliquity, eccentricity and precession (orbital longitude of the perihelion) that respectively describe the tilt of Earth's rotational axis, the shape of the Earth's orbit and the seasonal timing of Earth's closest proximity to the Sun. Our common view of seasonality imply that seasonal cycles occur because the Earth's axis is tilted, with seasons having opposite timing in the two hemispheres. At low latitudes, where the surface inclination experiences limited annual changes, the seasons are relatively mild. In some cases, such as on the

Equator, seasonal variations in climate are barely noticeable except for changes in humidity, as for the monsoon. In the open ocean, the modern seasonal thermal contrast corresponding to the annual amplitude of Sea Surface Temperature (SST) is in many tropical places much less than 2°C (e.g. Levitus, 1982; Harrison et al., 2009; Erb et al., 2015; Chiang and Broccoli, 2023).

While this likely hold for present-day climate, the annual amplitude of SST at the equator might have been different in the past during periods of high eccentricity (e = 0.050-0.060), owing to an annual variation of Earth-Sun distance of about 10-11% - compared to 3.2 % under present day orbital configuration (e = 0.016) using the ellipse equation (Figure 1). At high eccentricity, Earth's more elliptical orbit indeed amplifies the effect of precession (Berger and Loutre, 1991). The latter alters the timing of the perihelion, with a periodicity of ca. 21 ka and determines the season of increased insolation. Elevated eccentricity

amplifies the difference between the energy received at low latitudes at perihelion versus aphelion, inducing potential tropical seasonal fluctuations that impact meteorological factors (Chiang and Broccoli, 2023). The impact of eccentricity-modulated precession variations has been extensively documented for various low latitude seasonal phenomena such as monsoons (e.g. Wang, 2009; Cheng et al., 2016; Prescott et al., 2019), ocean primary productivity (Beaufort et al., 1997; Le Mézo et al., 2017), African lake levels (Trauth et al., 2009), the Dole effect (low latitude vegetation and phytoplankton productivity) (Landais et al.,

2010), and El Niño-Southern Oscillation (ENSO) (Clement et al., 1999; Timmermann et al., 2007; Ashkenazy et al., 2010; Erb et al., 2015). Those studies generally focused on specific intervals, preventing a systematic understanding of the processes at play at the scale of one full orbital precession cycle as well as of the eccentricity dependence of the system response.

       Very few studies focus on trying to understand the dependence of the climate system's response to eccentricity. Erb et al.

(2015) simulations clearly exhibit a strong amplification of seasonal cycle under extreme eccentricity compare to null eccentricity in the east Equatorial Pacific region, but the implication of their results for the understanding of eccentricity effect on seasonality was not revealed by that time. The study indeed mostly focused on precession and obliquity forcing, providing in-depth analysis on the dynamical mechanisms at play in the response of the regional surface ocean temperature. Building on Erb et al. (2015) simulation results Chiang et al. (2022) more recently showed that the increase in seasonal SST amplitude

under high eccentricity in the Pacific cold tongue was a robust phenomenon. Their analysis highlights that the SST seasonal cycle in the cold tongue region results from the combination of two distinct cycles that are respectively driven by the tilt of the Earth axis (i.e. tilt effect) and the distance to the sun (i.e. distance effect) both generating distinct seasonal pattern. They further show that the distance effect equals the tilt effect under extreme eccentricity configuration (e= 0.05), but only contributes to 1/3 of seasonal amplitude under present-day orbital forcing ( 0.4°C) which might explain why it remains ignored. Chiang and

Broccoli (2023) further expand the concept of the distance effect at global scale showing that under present-day eccentricity the latter is contributing to above 20 % of seasonal amplitude of surface temperature in most of tropical areas but also of precipitation and winds in both the tropics and extra-tropics. The concept of eccentricity-driven amplification of seasonality might extend beyond just the "physical" aspects of the climate system to include processes like primary productivity in the ocean. Primary productivity integrates signals from temperature, the hydrological cycle, and wind circulation, all of which play a role

 in shaping the surface ocean environment (e.g. Beaufort et al., 1997).

Beaufort et al. (2022) also investigated the direct impact of Earth's orbital eccentricity on tropical surface ocean seasonality building on paleoceanographic records from the Indo-Pacific Warm Pool. The cyclic diversification phases they observed in the evolution of coccolithophores spanning 2.8 million years closely follow the heightened seasonality during periods of high eccentricity. The distinctive eccentricity signature found in the records indeed differed from the one of the Pleistocene global climate cycles and ice volume variability, which are rather following high latitudes insolation forcing. While this study provided evidence of a significant seasonal pattern in surface conditions of the Indo-Pacific Ocean basins under high eccentricity, the simulation design they used to support their hypothesis only had two precession configurations and therefore felt short at capturing an entire precession cycle at high eccentricity, limiting a comprehensive analysis. Apart from the aforementioned investigations, only a limited number of modeling studies have addressed the combined sensitivity of SST and Primary Production (PP) dynamics to eccentricity. These factors are nevertheless crucial in assessing paleoclimate dynamics, given the numerous proxy records associated with them.

In this study we used an Earth System Model that includes a marine biogeochemistry module to simulate both SST and PP response to changes in precession at high eccentricity that we compare with low eccentricity configurations (Beaufort et al., 2021; Sarr and Beaufort, 2024). Given the complex response of ocean surface to precession forcing at regional scale (Erb et al., 2015; Chiang et al., 2022; Beaufort et al., 2001) a full precession cycle is necessary to understand the long-term seasonal dynamics associated with the timing of perihelion. Our study is based on eight simulations: at high eccentricity, a precession cycle is described by six different longitudes of the perihelion, while at low eccentricity, two longitudes of the perihelion are sufficient, as precession has a limited effect when the orbit is nearly circular. Our setup aims at identifying the direct ocean response to changes in insolation, excluding the potentially competing effects of change in ice-sheet extent, $pCO_2$ or nutrient supply via runoff or dust. Our results show that at high eccentricity, the low latitudes ocean experienced significant seasons, in SST and PP, related to a stronger annual change in the Earth-Sun distance, confirming previous studies focusing on the East Equatorial Pacific cold tongue (e.g. Erb et al., 2015; Chiang et al., 2022; Chiang and Broccoli, 2023) or on the Indo-Pacific Warm Pool (Beaufort et al., 2022).

## 2 Methods

### 2.1 Model and simulations setup

We used the IPSL-CM5A2 Earth System Model that integrates three key components: the LMDz5A atmospheric model (Hourdin et al., 2013), the ORCHIDEE land surface model(Krinner et al., 2005), and the NEMOv3.6 oceanic model (Madec and the NEMO team, 2015). The NEMO model encompasses the ocean dynamics component (OPA : Madec, 2008), a sea-ice thermodynamics model (LIM2 : Fichefet and Maqueda, 1997), and a biogeochemistry model (PISCES-v2 : Aumont et al., 2015). The ocean grid has a horizontal resolution of 2° by 2° (refined to 0.5° in the tropics) and 31 vertical levels, with varying thickness

from 10 m at the surface to 500 m at the ocean floor. The atmospheric grid has a horizontal resolution of 1.875° in latitude by 3.75° in longitude and incorporates 39 vertical levels. The OASIS coupler (Valcke et al., 2012) facilitates the ocean-atmosphere coupling by interpolating and exchanging variables between the two components. A detailed description of the IPSL-CM5A2 model and its performance in simulating pre-industrial climate and ocean can be found in Dufresne et al. (2013) and Sepulchre et al. (2020).

The ocean biogeochemistry component of the model, PISCES-v2 (Aumont et al., 2015), simulates the primary oceanic bio-geochemical cycles (C, P, Si, N, and Fe) and includes a simplified representation of lower trophic levels within the marine ecosystem. It incorporates two phytoplankton size classes (nannophytoplankton and diatoms) and two zooplankton size classes (micro- and meso-zooplankton), along with five limiting nutrients (Fe, $NO_3^-$, $NH_4^+$, Si, and $PO_4^{3-}$). Phytoplankton growth is influenced by nutrient availability, light penetration, and water temperature. In the version of the model we used, river supply of all elements except DIC and alkalinity remains constant across simulations and is obtained from the GLOBAL-NEWS2 datasets (Mayorga et al., 2010). For further insights into the model parameterizations, see Aumont et al. (2015).

We conducted eight equilibrium simulations, distinguished solely by their respective orbital parameters as shown in Table 1. Four of the simulations that we are using here have already been introduced in Beaufort et al. (2022) and Beaufort et al. (2021). Out of the eight simulations, six were performed at high eccentricity, representing the most extreme values observed during the past million years. These high eccentricity simulations encompassed six distinct angles of precession in order to achieve a 60° resolution ( 2 months) of the perihelion motion in the orbital plane during a precession cycle. Additionally, two simulations were carried out at the lowest eccentricity, where only two precession angles were considered due to the negligible impact of precession when eccentricity is low. Land-sea mask, ice-sheets configuration, as well as concentrations of $CO_2$ and other greenhouse gases were all set to pre-industrial values so we only focus on the direct effect of orbital configuration on the surface ocean. Nutrient supply by the rivers is also kept constant from one simulations to the other. Each simulation was initiated from an equilibrated pre-industrial simulation conducted by Sepulchre et al. (2020) and was run for 500 additional model years.

In the following, all variables are displayed as monthly averages over the final 100 years of each simulation and the net primary productivity is calculated by integrating values over the entire water column. We acknowledge that the duration of seasons is governed by the timing of perihelion, consequently influencing the length of each month in the Gregorian calendar (Joussaume and Braconnot, 1997). Nevertheless, in the context of this study, the impact remains limited, except for instances where monthly alignment to the calendar is presented. In order to avoid this problem, we do not adopt an angular definition of month but the classical one in which the number of day per month does not vary with the precession (see Joussaume and Braconnot, 1997). The description of the Earth's orbital system is from Berger and Loutre (1991) and Laskar (2020) and is illustrated in Figure 1. The timing of the orbital solution is from Laskar et al. (2004).

## 2.2 Comparison with modern conditions

For comparison with simulations, SST and PP datasets were used to illustrate their seasonality and annual average values. Modern SST data were gathered from the Comprehensive Ocean-Atmosphere Data Set (COADS), which compiles marine observations conducted by ships of opportunity between 1854 and 1992. These SSTs are edited and statistically summarized on a monthly basis for the period of 1961-1992, utilizing 2° in latitude by 2° in longitude cells (Slutz et al., 1985). Modern PP data were acquired from MODIS satellite measurements taken between 1998 and 2021 (Kulk et al., 2020). This dataset provides monthly averaged measurements in grid cells of 7 km$^2$. The data are averaged at a resolution of 1° in latitude by 1° in longitude, and subsequently, monthly averages have been computed over the entire 1998-2021 period.

## 2.3 Seasonality analysis

In the following analysis we adopt the basic method of calculating the annual amplitude (e.g. Chen and Yu, 2015) of SST or PP by determining the difference between the highest and lowest monthly values for each grid point on the dataset, as in (Beaufort et al., 2022). We preferred this approach to seasonality indices commonly employed in the field of hydrology sciences to characterize the annual patterns of humidity or river flow - such as the duration of the rainy season, the seasonal ratio, or more intricate methods involving complex histograms (e.g. Laaha and Blöschl, 2006) - for its simplicity and robustness.

## 2.4 Annual mean and amplitude analysis

It is important to be able to quantify how eccentricity increases the ocean seasonality at low latitudes. One way to quantify this is to produce a map showing the difference in annual amplitude of SST and PP between high and low eccentricity conditions. It is evident that the nature of the ocean's response to changes in energy input varies depending on local oceanographic contexts, owing the dynamic response of the system generated by the orbital forcing (Chiang and Broccoli, 2023). For example, the solar declination during perihelion will greatly influence SSTs depending on latitude, with significant impact on the past dynamics of the El Niño-Southern Oscillation (ENSO) (Erb et al., 2015).We acknowledge that local dynamics may no longer be apparent in some peculiar areas such as the Cold Tongue or upwelling zones once the average of the six high eccentricity simulation are taken. However, the objective here is to assess the significance of eccentricity in the tropical ocean on a global scale. We however found that using an average of the simulation still highlight areas where changes in seasonality are significant, especially in the open ocean (Figure 3A). We also discuss comparison between the low eccentricity simulation versus single high eccentricity simulation (for 3 peculiar configuration of the perihelion) (eg. Figure 3B-D).

As the annual mean patterns of SST and PP are nearly identical between the two low eccentricity simulations (R$^2$>0.99, N=79932, Figure 4A and C), related to the reduced effect of precession at low eccentricity, we select only one of them (Ecc. min - P310) for comparison with the high eccentricity simulations in order to simplify our analysis.

## 3 Results

### 3.1 Sea Surface Temperature dynamics

#### 3.1.1 Mean annual SST

The simulated annual mean SST displays sufficient similarities between the different simulations and with modern measurements (COADS) (Figure 4A-C) to suggest with confidence that the annual mean is minimally affected by eccentricity. As shown on Figure 4, the simulations with perihelion in December and in August at low eccentricity (Fig. 4A for SST and 4C for PP) indeed appear largely better correlated than equivalent simulation at high eccentricity (Fig. 4B for SST and 4D for PP), as variation in the amount yearly radiative energy received by Earth in response to change in precession at low eccentricity is negligible (Imbrie et al., 1993, and Table 1).

#### 3.1.2 Annual amplitude of SST

SST seasonality, as represented by the annual amplitude of SST (Figure 3D-F), shows notable differences between different eccentricity states. In the present-day ocean and in the low eccentricity configuration, extensive ocean regions near the equator exhibit small seasonal amplitude (<2°C) (Figure 3D-E). In high eccentricity simulations, only a small region in the western Pacific displays, amplitudes lower than 2°C on average (Figure 3F). Despite the strong hemispheric pattern in the annual amplitude temperature response (Figure 5B-D) related to declination of the perihelion, we note that in regions where strong changes in seasonality are identified in Figure 5A (eg. Equatorial Pacific and Atlantic, northern Indian ocean), seasonality is higher under-high eccentricity no matter the longitude of the perihelion (Figure 5B-D). The contrast in seasonal amplitude between high and low eccentricity simulations along a full precession cycle, clearly indicates a substantial increase (at least 1°C on average) in seasonality across large part of the Indian, Atlantic, and eastern Pacific Oceans. It is worth noting that the observed increase in SST seasonality depicted in Fig 5A aligns with the simulation conducted by Chiang and Brocoli (2023, Fig. 6A of their paper), which illustrates the relative contributions of Earth-Sun distance and tilt to the annual cycle of surface temperature. Few grid points (1.6%), mostly located along the California peninsula, show a systematic lower amplitude under high eccentricity conditions compared to low eccentricity scenarios (Figure 5). The largest increase in seasonality with eccentricity occurs near the equator, (in average two time larger between 5°N and 5°S (1.2°C) compared to 0.6°C between both 30°N and 5°N and 5°S and 30°S).

#### 3.1.3 Annual temperature cycles

Figure 2 clearly demonstrates that the warm season, along with its latitudinal position, coincides with the timing of perihelion and its associated declination. This correlation underscores the direct and immediate impact of solar radiation intensity and position on Sea Surface Temperature (SST) at low latitudes during periods of high eccentricity. Additionally, the amplification of insolation variations during high eccentricity further enhances seasonality. Conversely, the regional heterogeneity observed in the signals depicted in Figures 3F and 5B-D reflects the dynamical effect associated with the forcing.

In line with the section on the annual amplitude of SST, the analysis of the tropical averaged seasonal cycle of SST (Figure 6) reveals dampen annual SST cycles in both the modern data and the low eccentricity simulations (<0.5°C), whereas the high eccentricity simulations display mean annual amplitude of approximately 2.2°C. In addition to the increase in SST seasonal amplitude at high eccentricity, the SST (averaged over 30°N-30°S) seasonal cycle strinkingly shift with precession. This happens only in the high eccentricity ensemble, in contrast to the modern and low eccentricity cases. Similar outcomes are observed when temperatures are averaged within a narrower latitude range (e.g., 5°N-5°S). The selection of this wide latitude range is intended to encompass tropical climate phenomena, including the monsoon on its oceanic area.

Interestingly, the timing of the temperature peak differs among the high eccentricity simulations, indicating a progressive displacement of the warmest period in the year following the shift in the longitude of perihelion ($\omega$) during a precession cycle (Figure 6). The warmest period typically occurs one to two months after perihelion, when the Earth reaches its closest distance to the Sun, whereas the coldest period occurs approximately one to two months after aphelion, when the Earth is farthest from the Sun. This establishes a direct connection between thermal seasons and precession, whereby the gradual but significant warming of the ocean surface is attributed to the increased radiative energy received from the Sun during perihelion. The spatial heterogeneity of the annual amplitude SST in maps shown in Figure 3, also suggests a redistribution of the signal. This is indicative that the thermodynamics effect is also distributed dynamically through winds and thermocline, similarly to what is described in (Erb et al., 2015) in the case of the equatorial Pacific. As expected, such a relationship between thermal seasons and precession is not observed in the low eccentricity cases (present-day and the low eccentricity simulation) ; despite their nearly opposite precession configurations ($\omega = 103°$ and $\omega = 310°$, respectively) both cases experience two relatively minor increases in temperature at the same periods of the year (May and October).

## 3.2 Primary Productivity dynamics

### 3.2.1 Mean annual primary production

The mean annual PP patterns exhibit strong similarity across all seven simulations (Figure 7A-C). First order patterns of PP are similar between present-day observations (MODIS dataset) and the simulations though simulated PP is under-estimated for the tropical Indian and Atlantic Oceans as well as in the western Pacific and overestimated in the East equatorial Pacific, as discussed by Aumont et al. (2015). However those small discrepancies do not affect our analysis that focuses on comparing high and low eccentricity simulations, as we expect the model to be biased in the same way.

Simulated PP align with observations, revealing an equatorial band with higher productivity, and lower productivity areas located in the tropical gyres. This equatorial high annual PP area is the result of Ekman upwelling that bring nutrient-enriched water to the sub-surface (e.g. McClain and Firestone, 1993). In the Indian Ocean for example, the high PP areas spread northward in the Arabian Sea and Bay of Bengal due to the Ekman dynamics forced by the monsoon (Bauer et al., 1991). The similarity between the two maps (Figures 7B, and 7C) suggests that mean annual PP is minimally affected by eccentricity.

### 3.2.2 Annual amplitude of Primary Production

The annual amplitude of PP exhibits heterogeneity and patchiness within the tropical band, as depicted in Figure 7. Areas characterized by seasonal upwelling, such as in the monsoon region, generally display the largest amplitudes, whereas oligotrophic regions exhibit smaller amplitudes. Simulations with high eccentricity often yield larger amplitudes compared to those with low eccentricity (Fig. 8). Approximately 70% of the area located between 30°N and 30°S exhibit an increase in seasonality, with an average 11% increase in PP amplitude at high eccentricity (Fig. 8A). The amplification of seasonal amplitude is particularly remarkable in the Indian Ocean and narrow equatorial bands of the western Pacific and Atlantic Oceans. The regions exhibiting the most pronounced change in PP amplitude between high and low eccentricities shift across the range of simulations (Fig. 8C-D). For instance, the Arabian sea upwelling region demonstrates the largest change in amplitude when the longitude of perihelion ($\omega$) is 87°, while the smallest change is simulated at $\omega$ = 315° (Fig. 7B, C). When $\omega$ = 210°, the Indo-Pacific Warm-Pool displays strong seasonality, while much of the rest of the tropical oceans (equatorial Atlantic, eastern Pacific, and monsoon area) exhibit decreased seasonality compare to the low eccentricity case (Fig. 8D).

The impact of eccentricity on the annual amplitude of PP is clearly illustrated in Figure 9, where the average PP and annual amplitude of PP between 30°N and 30°S are plotted for each simulation. The results reveal that the annual amplitude of PP significantly rises by up to 19% on average at low latitudes during periods of high eccentricity compared to low eccentricity, except when the longitude of the perihelion occurs in May. In contrast, increasing eccentricity does not seem to have a significant impact on mean annual PP, as indicated by the minimal fluctuation in the simulated PP at high eccentricity (+/- 2% on average compare to the simulated PP at low eccentricity, Fig. 9A).

### 3.2.3 Seasonal cycles of primary production

Unlike SSTs in the tropics, which lack strong seasonality under low eccentricity conditions, tropical PP exhibit amplified seasonality due to the seasonal variation in declination. This seasonality is seen as partly due to strongly seasonal phenomena associated to the migration of the Intertropical Convergence Zone (ITCZ) and monsoons(e.g. Longhurst, 1995). The annual cycle of PP remains relatively consistent across different precession states, but is significantly amplified in simulations with high eccentricity (Figure 10). Two seasons of lower averaged PP (when averaged over 30°S-30°S) following the equinoxes are depicted in the seven simulations and modern observations. The highest tropical PP generally occur two months after the solstice.

The impact of eccentricity amplification can be examined by plotting PP in the precession time domain instead of the traditional annual time model (Figure 11). This representation reveals that the peak PP occurs when the perihelion aligns with a particular month or slightly earlier, and conversely at any given month, low PP lower occurs after the aphelion. It is important to note that each month in this figure is scaled differently, with the highest scale assigned to August because it is the month showing the highest PP in all simulations and modern data. Remarkably, the maximum productivity, reaching approximately

220 gC/m$^2$/year, coincides with the perihelion occurring in August ($\omega$=315°). This demonstrates that the regular shift of the perihelion during a precession cycle is intensified under high eccentricity conditions, similar to the impact observed on SST.

## 4   Discussion

### 4.1   Increased seasonality at high eccentricity

#### 4.1.1   Comparison with previous studies

Our simulations exhibits an amplification of temperature seasonal cycle in the tropical area-average ocean during periods of high eccentricity of Earth's orbit. This effect is particularly evident for the SSTs, especially in the open ocean (Fig. 3D-E) for which the seasonal cycle is almost muted in the present-day observations and low eccentricity simulations, with an annual temperature amplitude of less than 0.5°C on average. In contrast, the high eccentricity simulations exhibit an annual thermal amplitude of 2.2°C on average, highlighting the significant increase in seasonal variability. This result is in agreement with Ashkenazy et al. (2010) who simulated equatorial (4°N-4°S) oceans at high eccentricity with the autumn and spring equinoctial precessions (201,000 and 213,000 years ago) and found large SST seasonal cycles of about 2°C in the three oceanic basins with limited change in mean annual temperature. Our results also align with Erb et al. (2015) simulations exhibiting an increase in SST seasonality by 2°C on average in the equatorial Pacific in the idealized high eccentricity cases (e = 0.049) compared to the preindustrial one (e= 0.0167, $\omega$=103°), while changes in annual mean SST remain relatively small. More importantly our simulations results are coherent with Chiang et al. (2022) and Chiang and Broccoli (2023) which recently highlight that eccentricity does affect seasonality of SST in the tropics, even if its role have remained under-estimated. Their physical analysis indeed show that while at present day the distance effect (ie. the change in Earth-sun distance during the year that depends on the eccentricity of Earth orbit) might only be responsible for up to 20 % of the temperature seasonal amplitude at present day, its contribution increases with increasing eccentricity, fostering an amplification of the seasonal cycle.

Averaging over different precession simulations at high eccentricity, as we did in this study, prevents us from discussing regional peculiarities, as done in Erb et al. (2015), Chiang et al. (2022) or Braconnot et al. (2008), and therefore from providing robust physical analysis. While we opt for this method to render the discussion of our results and their implication for transient paleoceanographic record interpretation easier, we acknowledge that additional in-depth investigations are required at more regional scale to better understand the dynamic of seasonality. Previous studies however show that the distance effect on seasonal amplitude combines a thermodynamics response to direct insolation with a dynamical response (Erb et al., 2015; Chiang et al., 2022; Chiang and Broccoli, 2023). Chiang and Broccoli (2023) further suggest that the dynamical response is driven by the thermal contrast between the Pacific basin (Marine Hemisphere) and it continental counter part centered on the African continent (Continental Hemisphere), resulting in zonal displacement of the Walker Circulation. In the Cold Tongue region this shift in Walker uplift region generates anomalous easterly winds that drive the dynamical response, similar to the

ENSO dynamics, albeit at different time scale (Chiang et al., 2022).

The seasonality of primary productivity is also increased during periods of high eccentricity (Beaufort et al., 2022). This is related to the fact that primary productivity in the tropical band is strongly influenced by wind dynamics, which are commonly assumed to be linked to the inter-hemispheric seasonal contrast in SSTs. For example in the northern Indian Ocean, present-day productivity is stronger during boreal summer when the strong south-westerlies associated with South Asian summer monsoon drive clockwise surface circulation that is responsible for upwelling of cold and nutrient enriched water to the surface, boosting

PP (e.g. Koné et al., 2009). Anomalous wind circulation under high eccentricity generate either stronger upwelling ($\omega = 315°$) or strong convective mixing ($\omega = 87°$) (Beaufort et al., 2022). Similar dynamical response with anomalous wind circulation impacting thermocline depth in the Eastern equatorial Pacific that could likely be related to PP signal are also described in the east Equatorial Pacific (Erb et al., 2015; Chiang et al., 2022). In addition to wind-driven response of surface ocean circulation, Beaufort et al. (2022) also highlight that the seasonal enhancement of PP under high eccentricity also respond to change in

hydrological cycle that modify local salinity or solar radiation received in the surface ocean creating more or less favorable condition for PP. This overall suggests that deeper understanding of mechanisms at play in PP response to increase in eccentricity requires more regional analysis.

### 4.1.2   Transient climate signal for data-model comparisons

To visually represent how eccentricity and precession affect SST and PP over time, we computed an approximation of a transient signal spanning a 900,000-year period (Figure 12). This is achieved by applying a linear scaling of eccentricity to the outputs of six precession simulations, which are arranged chronologically based on known precession and eccentricity values (Laskar et al., 2004). The series are based on the difference between the six simulations that describe a precession cycle with a temporal resolution of about 3.8 thousand years (a sixth of a precession cycle) at an eccentricity of 0.054 and the values of the

low eccentricity simulation. By employing proportional reasoning, we can then scale the values of that difference to account for other eccentricities at times given by the longitude of perihelion. In other words, the six values are being repeated changing only according to the eccentricity at time $t$. The resulting time series will be expressed in the following equation (1):

$$Yt = (HE_\omega - LE).e_t/(0.053 - 0.005) \quad (1)$$

Here, $Yt$ represents annual average or amplitude at time $t$ for SST or PP, $HE_\omega$ denotes the simulated value for a given perihelion longitude ($\omega$) under high eccentricity, $LE$ signifies the simulated value under low eccentricity ($\omega= 310°$), $e_t$ represents the eccentricity at time $t$, and 0.053 and 0.005 correspond to the eccentricity values used for the high and low eccentricity simulations respectively. This equation generates time series representing the estimated Sea Surface Temperature (SST) or Primary

Productivity (PP) across tropical regions as illustrated in Figure 12. These series offer a temporal resolution of approximately 3.8 thousand years. The equation assumes a linear correlation between seasonality and eccentricity, although this relationship

has yet to be empirically validated. It is important to note that these time series do not aim to mirror real-world conditions but rather provide scaled values influenced by the dynamics of eccentricity and precession, while other parameters like greenhouse gases, ice volume and sea level are held constant.

Figure 12 illustrates that while the mean annual SST signal in the tropics remains unaffected by eccentricity, the seasonal amplitude experiences a significant increase at eccentricity frequencies (ca. 100 ka and 400 ka) while precession cycles are hardly visible. On the other hand, annual mean PP displays a higher sensitivity to precession than SST, while its response to eccentricity seems to be dampen. Eccentricity does however significantly affect PP seasonality, except when the perihelion aligns closely with the vernal equinox. Unlike SST, PP exhibits inherent seasonality in the modern low latitude ocean. For instance, in the Indian monsoon region, PP reaches its peak during summer and declines at the solstices (e.g. Longhurst, 1995). This because seasonal dynamics of productivity in those regions is strongly tied to the oceanic circulation associated with ITCZ that crosses the equator twice a year during solstices (e.g. Longhurst, 1995; Pennington et al., 2006). This occurs because wind-driven open ocean upwelling (e.g. Mann and Lazier, 1996) reaches a maximum when the ITCZ has seasonally migrated farthest from the equator, aligning with the peak of the summer seasons in both hemispheres (Longhurst, 1995): the winds are minimal at the atmospheric convergence zones, including the ITCZ (Pennington et al., 2006). When eccentricity is high, this phenomenon is enhanced, preserving its seasonality. In other words, the PP phenology remains stable and tied to the calendar. Consequently, when the perihelion aligns with the equinox, PP annual mean also reaches its lowest values regardless of eccentricity levels. This explains why, in Figure 12, PP seasonality exhibits significant precession variability, in contrast to SST.

## 4.2 Gradual Drift of Seasons within the Calendar Year

Figure 2 shows that the seasonal cycle of SSTs is influenced by the annual positioning of perihelion and aphelion, with a lag of a few months. This result is inline with previous studies investigating seasonality response to eccentricity modulated precession in the Equatorial Pacific . Clement et al. (1999) for example used a simplified ocean-atmosphere model to investigate the influence of precession cycles spanning 150,000 years on El Niño-Southern Oscillation (ENSO). They noted shifts in seasonal energy distribution across the tropical Pacific Ocean, with the strength and frequency of El Niño events impacted by the interplay of precession and eccentricity-driven changes in energy excess timing and location. This is coherent with previous observation of a gradual shift of the warm SST period over a precession revolution at high eccentricity simulated in the East equatorial Pacific (cold tongue) (Erb et al., 2015; Chiang et al., 2022).

In our study, the six simulations we performed at high eccentricity allowed us to observe a complete revolution of precession at a 2-month resolution in all tropical oceans (Figures 6 and 11), highlighting a gradual shift of seasons (Fig. 13A). This gradual shift of seasons, which forms the basis of the precession description, has already been used to describe past dynamics of important low latitude phenomena such as monsoons (Braconnot et al., 2008) and ENSO (Clement et al., 1999; Erb et al., 2015; Chiang et al., 2022). Our study expands the scope, demonstrating that this phenomenon has a strong impact on the oceans

in the entire tropical band. Additionally, we want to stress that the seasonality pattern described here differs from the familiar seasons experienced at mid-latitudes, where summers and winters are defined to start at the solstices. Our results rather suggest that in low latitudes and at high eccentricity, the warm season does not begin at the same calendar date each year and rather progresses along the Earth's orbit and calendar at a rate of approximately 0.017° per year ( 25 minutes), equivalent to a cycle of approximately 21,000 years. This advancement occurs due to the shifting moment when the Earth approaches its closest point to the Sun (perihelion), attributed to the precession of the equinoxes.

We investigated whether the 'tropical seasons,' as described earlier, exert an influence on PP, which typically follows also a 'classical' seasonal cycle and the migration of the ITCZ, dictated by the tilt of the Earth's axis. Figure 13B illustrates the difference in the monthly average PP values over 30°N-30°S between the low eccentricity simulation and each precession configuration at high eccentricity for the corresponding month. The plotted results mirror those in Figure 13A, demonstrating a shift in PP similar to SST but with a reduced delay from perihelion. These findings emphasize the significance of the 'tropical seasons' on PP dynamics. The seasonal variation in radiative energy at high eccentricity leads to a direct forcing on PP, as evidenced by its observed seasonality.

## 4.3 Eccentriseasons

Due to the distinct origin of the seasonal variations that we have illustrated in this study, arising from the Earth's orbital eccentricity rather than the tilt of its rotation axis, we suggest that a distinct and appropriate nomenclature is needed. This would help deciphering between the described phenomenon and the 'classical' view of the seasons and therefore 'putting back eccentricity in seasons' as called for by Chiang and Broccoli (2023). In addition a distinct nomenclature would benefit to future work as it would make more evident to identify and discuss the effect of eccentricity on seasons. We propose the term 'eccentriseason', derived from a clipped compound of 'eccentricity' and 'season'. Eccentriseasons are defined as seasons occurring at low latitude in response to the cycles of the Earth-Sun distance: their annual amplitude increases with eccentricity, and their timing gradually shifts of about 25 minutes per year on the calendar in accordance with the precession of the equinoxes (see Figure 1). Eccentriseasons are therefore distinct from the familiar extra-tropical seasons which remains stable in the calendar and are less dependent on eccentricity.

## 4.4 Implications of eccentriseasons for paleoclimatology

The increase in seasonality within the tropical ocean would significantly affect low latitude climate phenomena: the rise in tropical SST during key seasons has a notable impact on energy transfer, influencing monsoons, migration of the Intertropical Convergence Zone, and ENSO dynamics. The relationships between these phenomena and the amplified seasonality are intricate and have already been explored in dedicated studies (Clement et al., 1999; Timmermann et al., 2007; Braconnot et al., 2008; Ashkenazy et al., 2010; Erb et al., 2015; Chiang et al., 2022). Our study does not focus on exploring these mechanisms but rather generalizes their impact on the tropical oceans, aiming to make them more accessible to the paleoceanographic

community.

The effect of high eccentricity on seasonality would also affect ocean's ecology, and in particular that of phytoplankton, that represents the cumulative outcome of localized climatic mechanisms. A study exploring phytoplankton's biological evolution has delineated the ecological impacts of eccentriseasons on marine phytoplankton (Beaufort et al., 2022): over the past 2.8 million years, the evolution of coccolithophores has been observed to directly align with the eccentricity cycles, displaying minimal influence from global climates. This pattern has been interpreted as a result of cyclic diversification in low latitude ecological niches during periods of heightened tropical seasonality in high eccentricity times. The present work highlights significant shift in SST seasonality, which likely plays a crucial role in the mechanisms (e.g. wind patterns, ocean circulation intensity, biologic productivity, biologic evolution) described in Beaufort et al. (2022).

The lack of a significant precession effect on low latitude mean annual primary production in our simulations is surprising, given that many paleoproductivity records show a strong response to precession (e.g. Molfino and McIntyre, 1990; Beaufort et al., 1997; Villanueva et al., 2001; Moreno et al., 2002; Su et al., 2015; Deik et al., 2017; Tangunan et al., 2017). Past PP reconstructions for coccolithophores rely on annual reconstructions of primary production (Beaufort et al., 1997; Hernández-Almeida et al., 2019). Modern data for primary production (Kulk et al., 2020) suggest that a linear relationship exists between mean annual PP and the annual amplitude of PP, with the latter being approximately half of annual PP (Fig. 14).

To further explored this, we compared our simulation results with a published record from the Indian Ocean (MD90-0963 core (Beaufort et al., 1997) for which the mean annual PP have been reconstructed using coccolithophore assemblages. We converted this annual PP record into seasonal amplitude of PP using a scaling factor of 0.59 that we calculated from the present day relation between annual mean and annual amplitude of PP in the Indian Ocean basin (Fig. 14). The comparison (Fig. 16) reveals strong similarities: the timing closely aligns, particularly between 720 ka and 570 ka, as expected due to the continued validity of the climatic explanation of PP dynamics provided in that paper (Beaufort et al., 1997). Furthermore, both the signal from the core and the one obtained from the simulations exhibit peak amplitudes of the same order of magnitude. This suggests that the significance of seasonality might have been overlooked when studying paleoproductivity in low latitudes.

Potential effect is also expected for SST proxies that are commonly used in paleoceanography because they are carried out by organisms that may therefore also have a preferred season of production. Alkenones and Mg/Ca are two proxies for SST that are commonly used (e.g. Brassell et al., 1986; Prahl et al., 1988; Rosell-Melé et al., 1995; Sonzogni et al., 1997; Rosenthal et al., 2000). The alkenones production presents a seasonality dependent on the coccolithophores *Emiliania huxleyi* phenology (Rosell-Melé et al., 1995; Sikes et al., 1997; Ternois et al., 1998). Similarly, planktonic foraminifera, produce tests from which Mg/Ca SST are estimated at specific seasons (e.g. Fairbanks et al., 1982; Mohtadi et al., 2009; Chaabane et al., 2023). The production seasons of these organisms has been used to explain differences in Holocene SST reconstructions from the same site using different proxies having different phenologies (Leduc et al., 2010; Lohmann et al., 2013; Bova et al., 2021). Our results suggest regular phenological phase shifts between primary production and sea surface temperature eccentriseasons.

For example, a plankton population always displaying peak production during the September high-productivity season would experience, throughout a precession cycle, a larger SST signal (typically with an amplitude of 2.5°C) than an organism that prospers year-round and is thus insensitive to the shift in the month of maximum SST related to eccentriseasons (Fig. 6). It is therefore imperative to account for the alignment between proxy producers' phenology and temperature fluctuations across the calendar year driven by eccentriseasons, ensuring accurate interpretations of SST records and mitigating seasonal biases. In line with our proposition, proxies accurately estimating mean annual SST (i.e., growing throughout the year) are expected to exhibit less precession-band variance that those with a seasonal growth pattern, which should record a precession component (Fig. 12).

An additional illustration of the impact of changes in annual SST amplitude can be observed in the evolution of phytoplankton in tropical regions, which responds notably to heightened seasonality during periods of high eccentricity, as discussed in (Beaufort et al., 2022). The morphologic evolution records presented in this work appeared to respond more to eccentricity than precession. This suggests that temperature plays a pivotal role in delineating distinct seasonal niches, thereby promoting diversification into new species among isolated phytoplankton populations.

## 5   Conclusions

In this study, we investigated the response of low-latitude surface ocean to variations in Earth's orbital eccentricity using the Earth System Model IPSL-CM5A2 and its marine biogeochemistry module PISCES-v2. Our climate simulations reveal that high eccentricity leads to increased seasonality in SST in low-latitude ocean surface waters, with an annual thermal amplitude of 2.2°C on average, in line with previous studies. In contrast, PP that already exhibits inherent seasonality under low eccentricity conditions, sees its seasonal cycle significantly enhanced under high eccentricity conditions. The consequences of this amplification of seasonality under high eccentricity configuration are significant: on long time scales, this would result in SST seasonality following only eccentricity frequencies, while PP seasonality follows both eccentricity and precession frequencies. The positioning of perihelion during the year directly affects the SST and PP seasonalities under high eccentricity, leading to a gradual shift of seasons within the calendar year. To account for this phenomenon, we introduce the term "eccentriseasons" that describes these distinct annual thermal differences observed in tropical oceans under high eccentricity conditions, which shift gradually throughout the calendar year. This study contributes to the growing body of knowledge about the role of orbital parameters in shaping Earth's climate over long time scales and highlights the significance of eccentricity-induced seasonality in low-latitude regions. Our results may hold significant implications for the understanding of low-latitude climatic phenomena with a strongly seasonal nature, such as ENSO and monsoons. We anticipate that this study will contribute to previous efforts by demonstrating the direct influence of Earth's orbital eccentricity on tropical seasonality both in paleoclimate modeling and data communities. The increased seasonality in tropical oceans under high eccentricity conditions can for example influence energy transfer and ocean dynamics which in turn affect those climate phenomena in the past. Our results also have potential implications for paleoclimatology studies: we highlight here the significance of seasonality as a parameter that may have been overlooked when studying paleoproductivity in low latitudes. This insight can inform the interpretation of paleoproductivity

records and proxies commonly used in tropical paleoceanography, because our study suggests that the interactions between seasonal production and shifting temperature seasonality likely imprints the signal that is recorded.

*Code availability.* Code availability LMDZ, XIOS, NEMO, and ORCHIDEE are released under the terms of the CeCILL license. OASIS-MCT is released under the terms of the Lesser GNU General Public License (LGPL). Up to date IPSL-CM5A2 source code is publicly available through svn, with the following commands line: svn co http://forge.ipsl.jussieu.fr/igcmg/svn/modipsl/branches/publications/IPSLCM5A2.1_11192019modipsl;cdmodipsl/util;./modelIPSLCM5A2.1 information regarding the different revisions used, namely:

- NEMOGCM branch `nemo_v3_6_STABLE` revision 6665
- XIOS2 branch `branches/xios-2.5` revision 1763
- IOIPSL/src svn `tags/v2_2_2`
- LMDZ5 branch `branches/IPSLCM5A2.1` rev 3591
- `branches/publications/ORCHIDEE_IPSLCM5A2.1.r5307` rev 6336
- OASIS3-MCT 2.0_branch (rev 4775 IPSL server)

The login/password combination requested at first use to download the ORCHIDEE component is anonymous/anonymous. We recommend that you refer to the project website: http://forge.ipsl.jussieu.fr/igcmg_doc/wiki/Doc/Config/IPSLCM5A2 for a proper installation and compilation of the environment. In addition, source code of the version used for this paper is publicly available at https://doi.org/10.5281/zenodo.6772699 (Pillot, 2022).

*Data availability.* The model outputs are archived at the SEANOE data repository : https://www.seanoe.org/data/00728/84031/ for the outputs published in Beaufort et al. (2022) and on a Zenodo repository for simulations specific to this study (https://doi.org/10.5281/zenodo.10951600) (Sarr & Beaufort, 2024)

*Author contributions.* LB and ACS designed the experimental approach. ACS performed the simulations. LB analyzed the results and prepared the figures. Both authors wrote the manuscript

*Competing interests.* LB is an editor at Climate of the Past

*Acknowledgements.* We thank Anthony Gramoullé for the help extracting relevant information from model outputs. We thank Christophe Menkes for helpful discussions, and Clara Bolton for reading an early version of the manuscript. Luc Beaufort is supported by the Agence Nationale de la Recherche (ANR22 ESDIR 003-ITCH). Anta-C. Sarr was supported by a grant from Labex OSUG (Investissements d'avenir – ANR10 LABX56) and was granted access to HPC resources of TGCC under allocation 2022-A0090102212 made by GENCI.

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

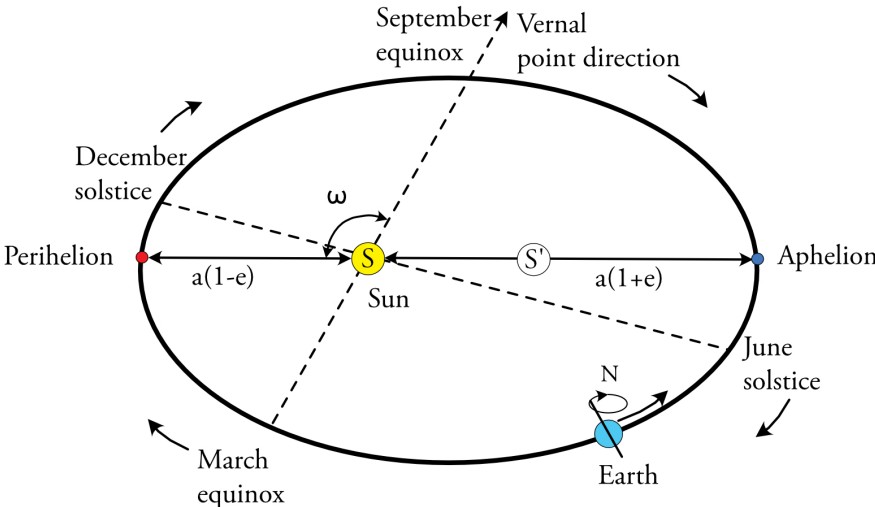

**Figure 1.** Schematic Illustration of Earth's Orbit Eccentricity and Precession (adapted from Berger et al., 1993; Laskar, 2020): The Sun (S) occupies one focus of the elliptical Earth orbit, which is traced counterclockwise. The eccentricity (e) signifies the ratio of the distance between the foci (S and S') to the major axis's length (2a). During perihelion, when Earth is closest to the Sun, the Earth-Sun distance equates to a(1-e). At aphelion, Earth's farthest point from the Sun, the distance becomes a(1+e). The annual Earth-Sun distance variation, in percentage, equals the eccentricity multiplied by 200. The perihelion longitude ($\omega$) denotes the angle between the Vernal Point direction (this direction is that of the Sun observed during the march equinox) and the perihelion directions. Precession causes a gradual clockwise shift of the Vernal equinox ( 25 minutes per year, or 60 minutes x 24 hours x 365 days / 21,000 years). The equinoxes and solstices are represented close to their modern positions where Earth is at the perihelion in early January.

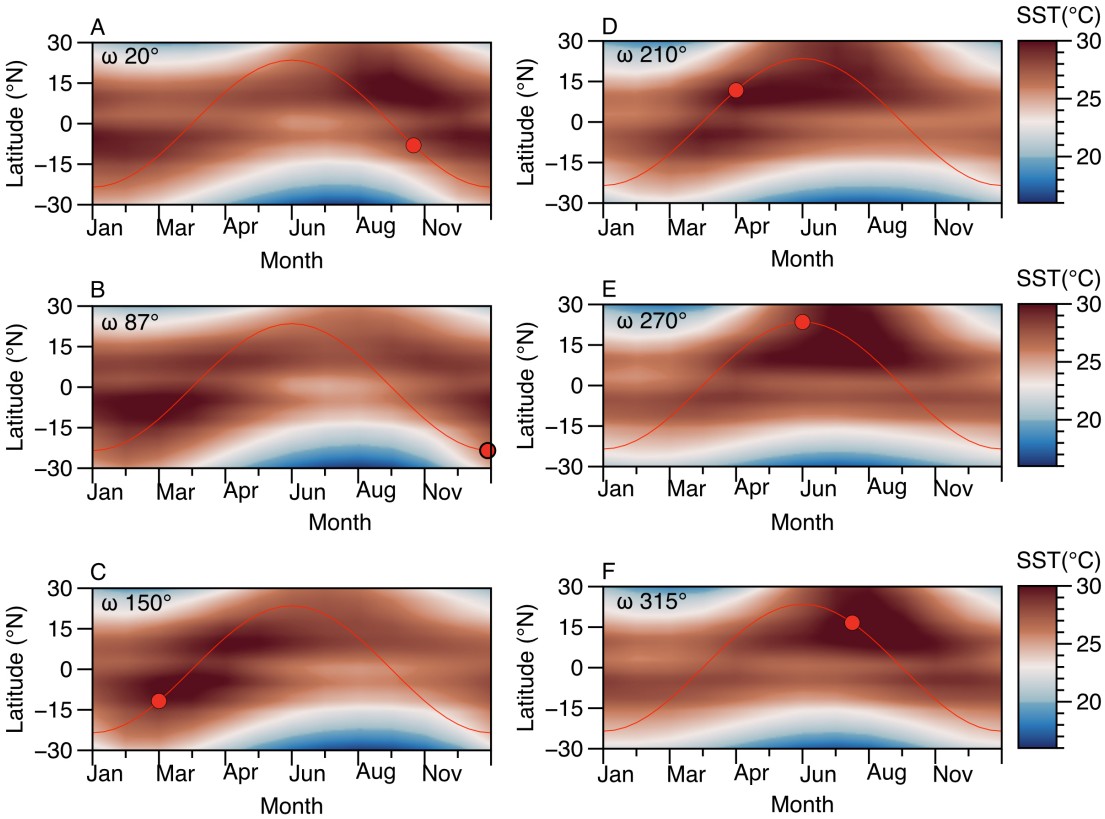

**Figure 2.** Significance of solar radiations on tropical SST during high eccentricity: Across six simulations featuring distinct longitude of the perihelion (A: $\omega = 20°$, B: $\omega = 87°$, C: $\omega = 150°$, D: $\omega = 210$, E: $\omega = 270°$ and D: $\omega = 315°$), SSTs are averaged across 1° latitudinal bands spanning from 30°S to 30°N over the course of the year. The red line depicts the seasonal declination of the Sun, while the red dot marks the declination of the Sun at the date of the perihelion.

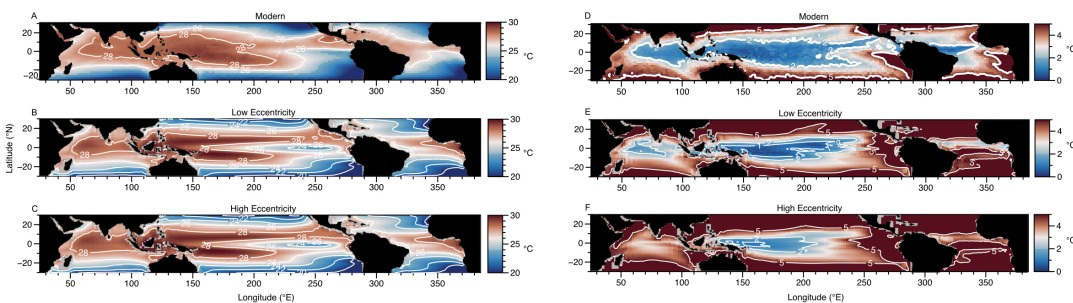

**Figure 3.** Sea surface temperature annual mean (A, B, C) and annual amplitude (D, E, F) from modern data (COADS) (A, D), from the simulation with low eccentricity ($\omega$ =310°) (B,E) and from the average of the 6 simulations with high eccentricity (C, F).

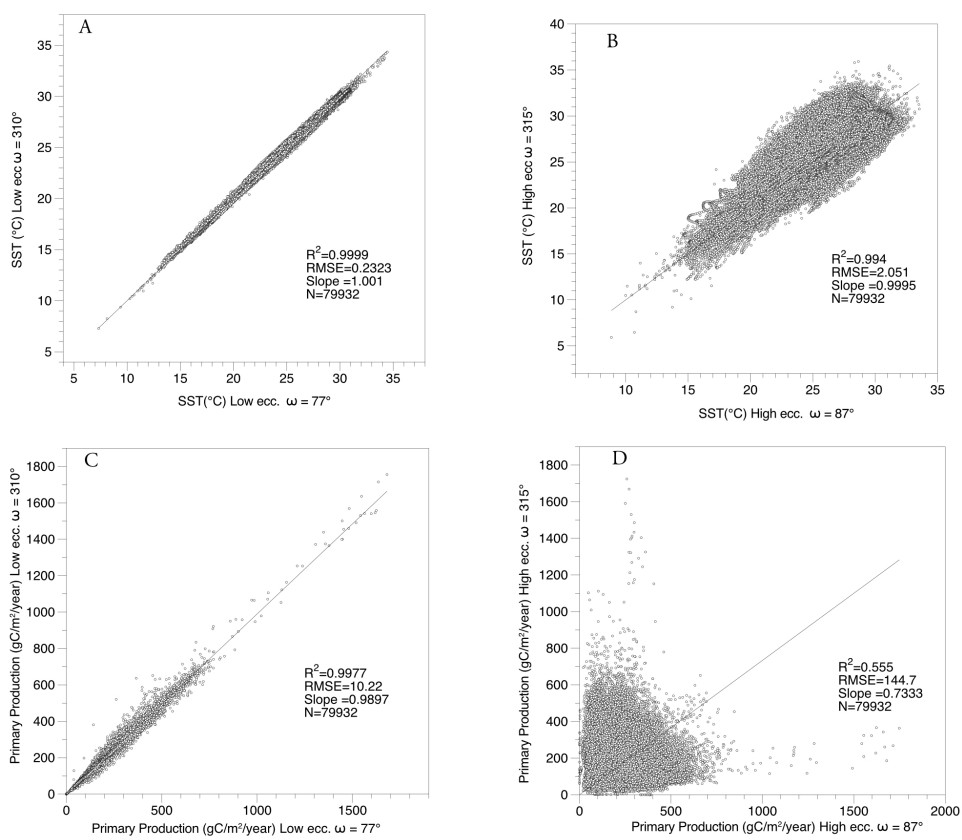

**Figure 4.** Correlation between simulations at opposite precession phases—low (A and C) and high eccentricity (B and D) for Sea Surface Temperature (SST) (A and B) and Primary Productivity (PP) (C and D) across the Tropical Ocean Band (30°N-30°S, 6661 pixels) Points are monthly average at one location. Each comparison is based on a total of 79,932 data points.

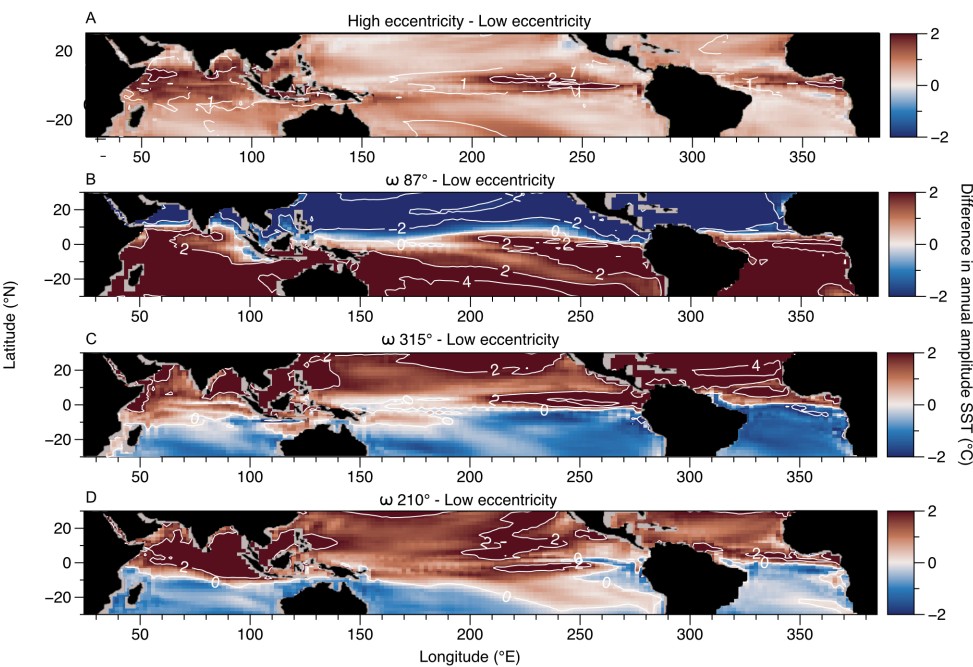

**Figure 5.** Difference of annual amplitude of SST between high (A: average of the 6 simulations; B : simulation with $\omega$ =87°; C: simulation with $\omega$ =315°; D: simulation with $\omega$ =210°) and low eccentricity simulation with $\omega$ =310°.

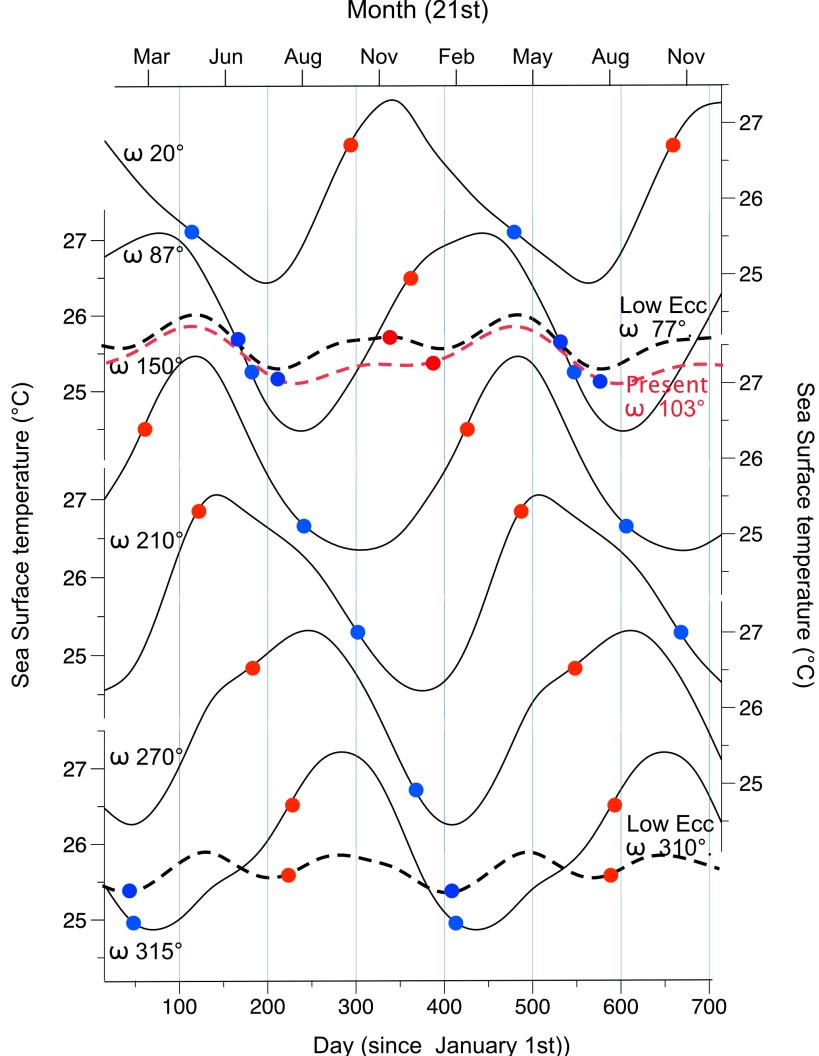

**Figure 6.** Two years time-series of SST averaged between 30°N and 30°S from high eccentricity simulations (solid lines), from low eccentricity simulations (black dotted line) and from modern data (red dotted line). The longitude of the perihelion ($\omega$) is indicated for each simulation. The red dots represent the date of perihelion, and the blue dots that of the apehelion.

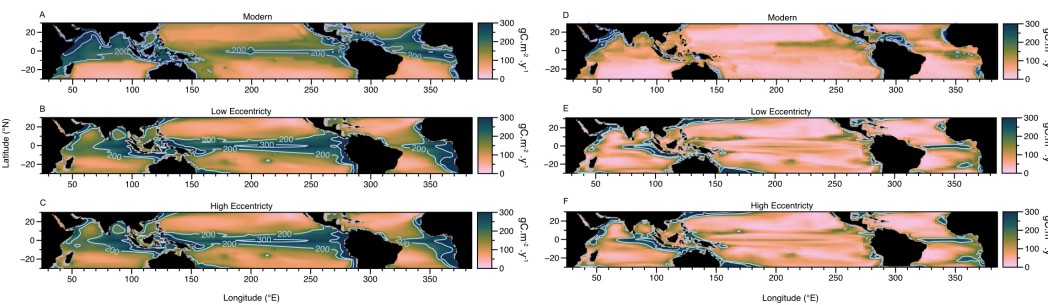

**Figure 7.** Primary production annual mean (A, B, C) and annual amplitude (D, E, F) from modern data (Satellite MODIS) (A, D), from the simulation with low eccentricity ($\omega$ =310°) (B,E) and from the average of the 6 simulations with high eccentricity (C, F).

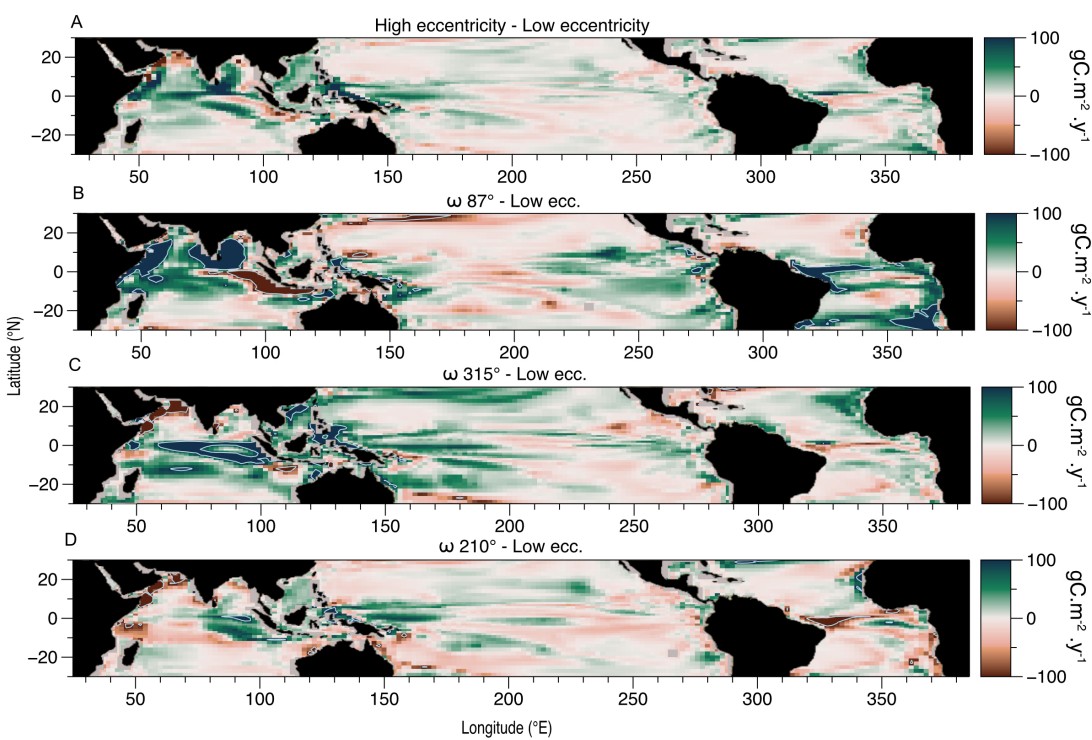

**Figure 8.** Difference of annual amplitude of primary production between high (A: average of the 6 simulations; B : simulation with $\omega$ =87°; C: simulation with $\omega$ =315°; D: simulation with $\omega$ =210°) and low eccentricity simulation with $\omega$ =310°.

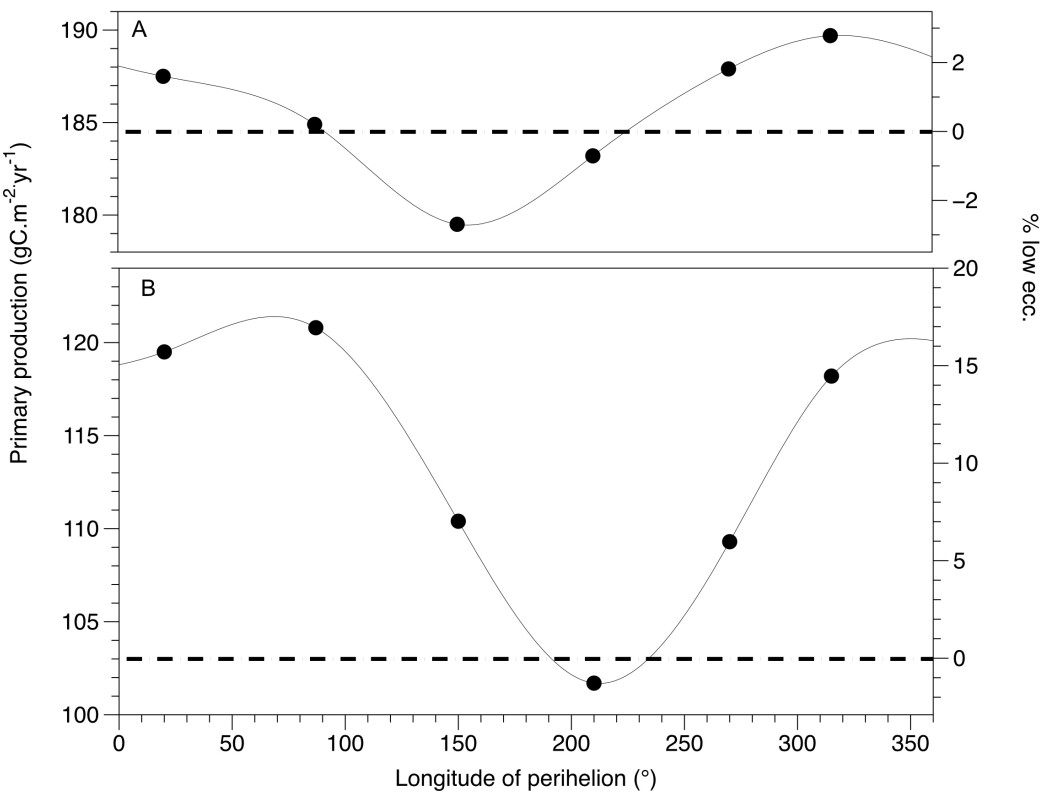

**Figure 9.** Evolution of primary production annual mean (A) and annual amplitude (B) averaged between 30°N and 30°S during a precession cycle for high eccentricity (solid line, each circle represent one simulation) and low eccentricity (dotted line), left scale in : gC.m-2.yr-1, right scale in percentage relative to low eccentricity simulation.

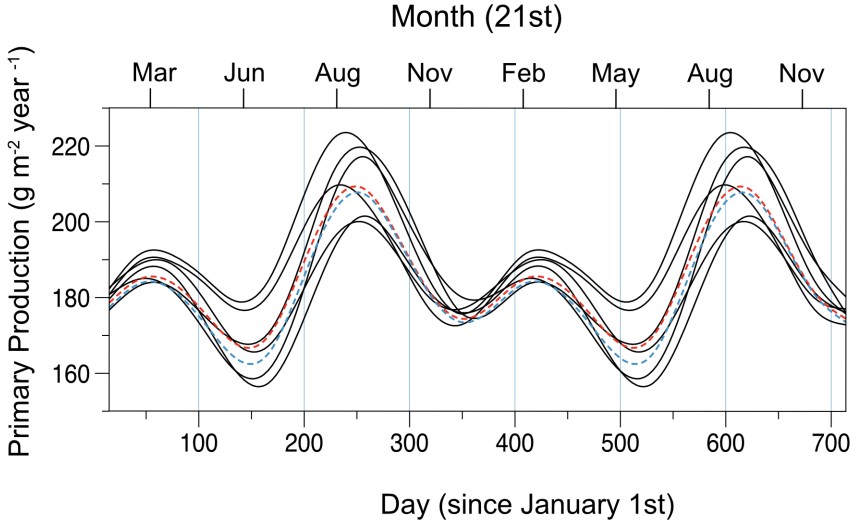

**Figure 10.** Two years time-series of primary production between 30°N and 30°S from high eccentricity simulations (solid lines). Low eccentricity simulations are represented by the red and blue dotted lines for $\omega$ =310° and $\omega$ =77° respectively.

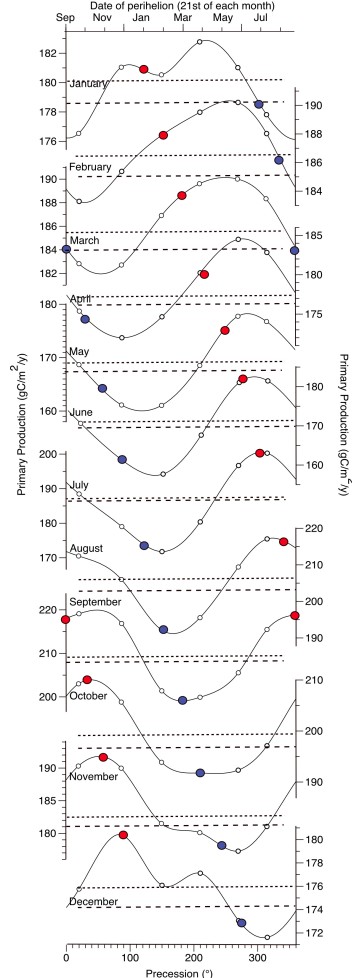

**Figure 11.** Monthly evolution of primary production at low latitudes (30°N-30°S) during a precession cycle simulated at high eccentricity (solid line) from January (top) to December (bottom). The values simulated by the low eccentricity experiments are represented by the dotted ($\omega$ =77°) and dashed lines ($\omega$ =310°) to express the range of variation during a precession cycle at low eccentricity. Color circles correspond to the position of the perihelion (red) and aphelion (blue) when it occurs in the same month.

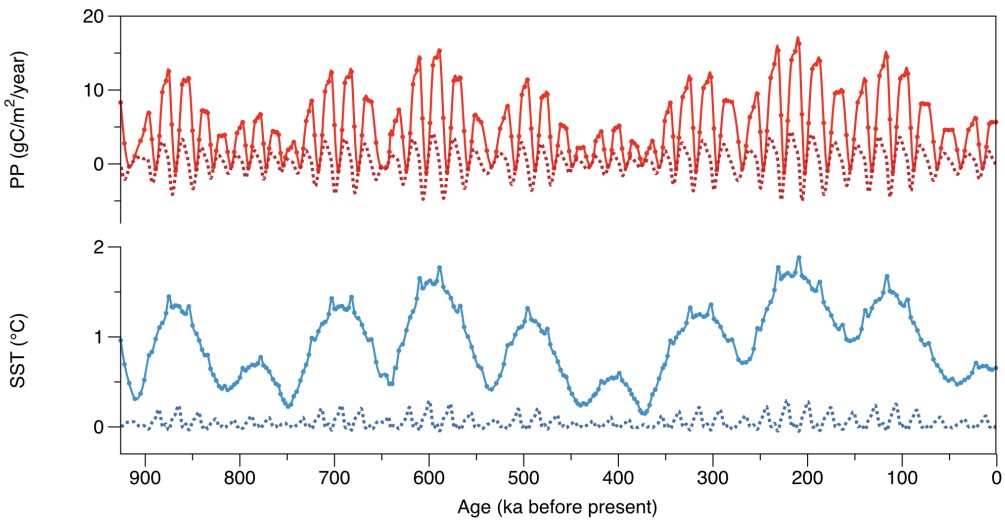

**Figure 12.** Transient signal of PP and SST over 900,000 years. The signal represent the response to precession and eccentricity forcing only (see text). The figure depicts SST (blue-bottom) and PP (red-top) annual mean (dotted lines) and annual amplitude (solid lines with circles). To obtain this signal, average values (30°N-30°S) for each of the six high eccentricity simulations are adjusted by subtracting the low eccentricity simulation ($\omega = 310°$) value. Obtained value are then arranged chronologically on a time frame based on precession Laskar et al. (2004) for the last 900,000 years. These values are further scaled by the eccentricity for each time point (see paragraph 4.1.2 for details).

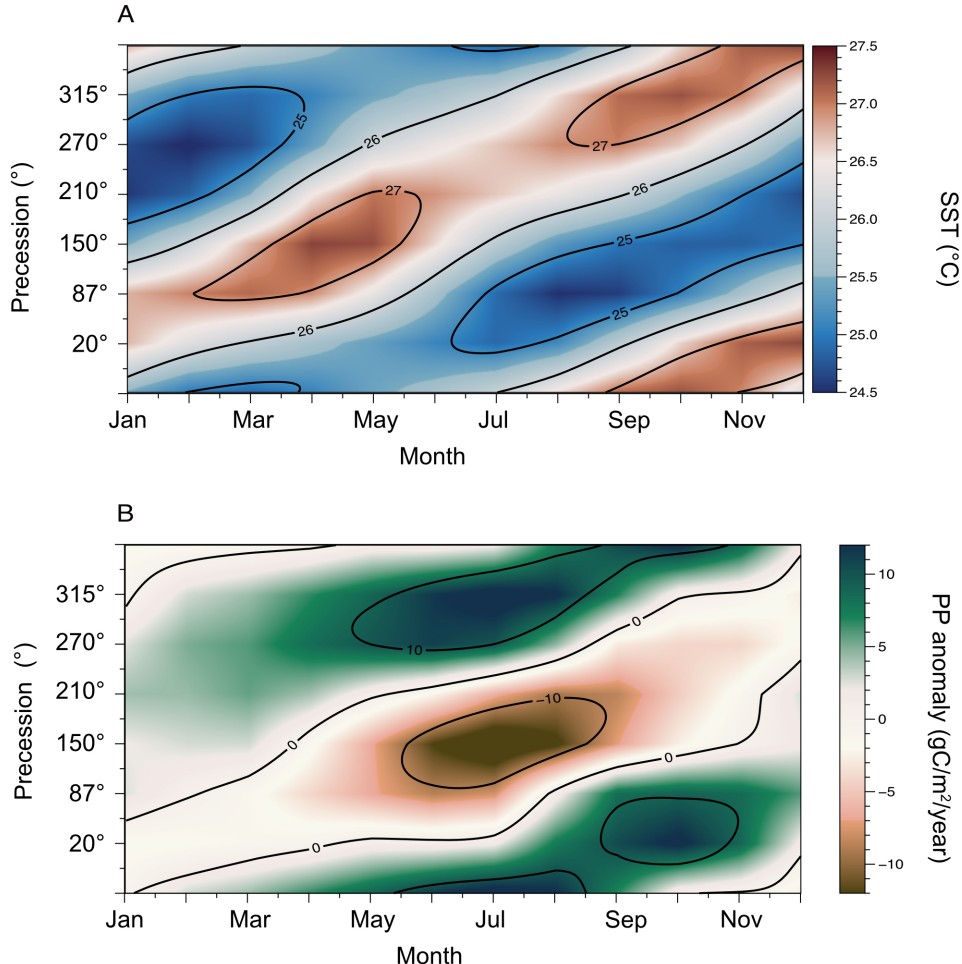

**Figure 13.** Annual variations of SST (A) and PP (B) averaged over low latitudes (30°N-30°S), along a precession cycle when eccentricity is high

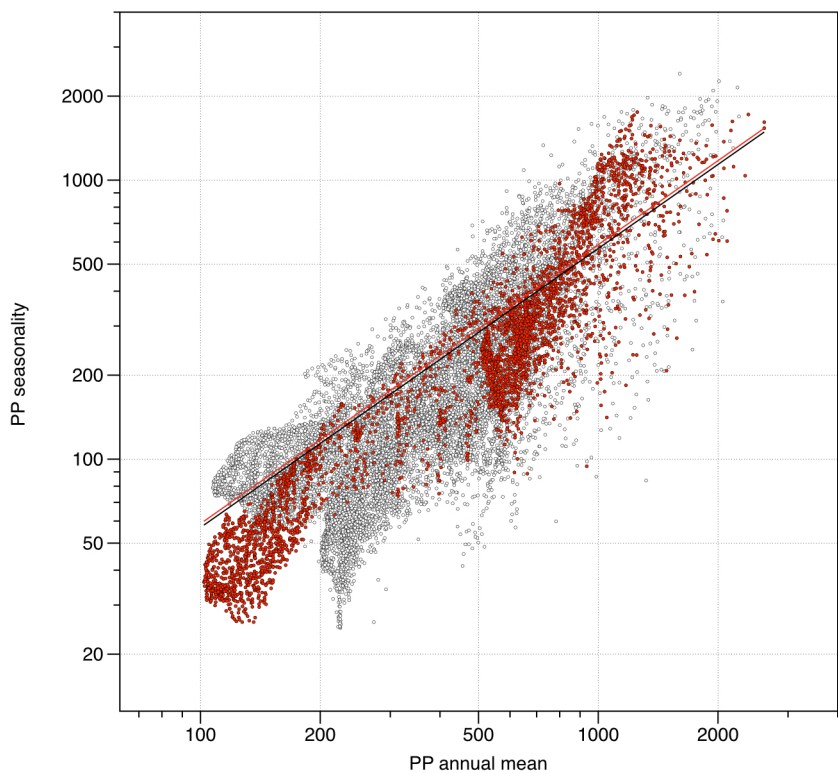

**Figure 14.** Relation between annual mean and annual amplitude (month max - month min) of primary production in the low latitudes (30°N-30°S) from MODIS satellite imagery Kulk et al. (2020). Black represent data for all oceans basin and red are values for the Indian ocean. Lines represent the intercept free regressions (All : N= 16981, $R^2$ = 0.82, slope = 0.57, Indian : N = 3532, $R^2$ = 0.85, slope =0.59))

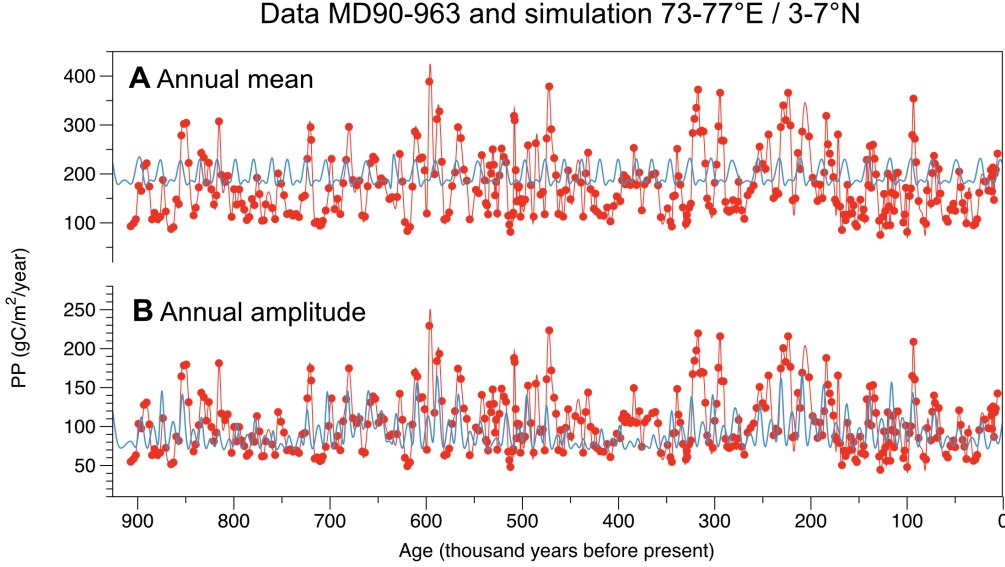

**Figure 15.** 900,000 years time series of annual (A) and amplitude (B) primary production in the Indian ocean at 75°E – 5°N (± 2°). Methods to construct the transient signal from the simulations (blue) are the same as figure 13 and described in paragraph 4.1.2. The estimated annual mean PP data (red) are from Core MD90-963 Beaufort et al. (1997). In B, the annual mean PP for the data have been scaled by 0.59 (see Figure 15 and paragraph 4.4) to represent the seasonal PP

**Table 1.** Summary of orbital parameters used for each simulation and annually averaged solar irradiance and its annual amplitude, both in W/m2

.

| Configuration name | Eccentricity | Longitude of perihelion (°) | Month at perihelion | Obliquity | Annually averaged solar irradiance | Annual amplitude | Reference |
|---|---|---|---|---|---|---|---|
| Ecc. min - P310 | 0.005 | 310 | August | 23.45 | 329.2 | 38.5 | Beaufort et al., 2022 |
| Ecc. min - P77 | 0.006 | 77 | December | 23.74 | 328.7 | 36.5 | Beaufort et al., 2022 |
| Ecc. max - P20 | 0.054 | 20 | October | 23.73 | 329.5 | 78.9 | This work |
| Ecc. max - P87 | 0.054 | 87 | December | 23.45 | 329.2 | 73.8 | Beaufort et al., 2022 |
| Ecc. max - P150 | 0.054 | 150 | March | 23.73 | 328.7 | 93.9 | This work |
| Ecc. max - P210 | 0.054 | 210 | May | 23.73 | 329.1 | 101.7 | This work |
| Ecc. max - P270 | 0.054 | 270 | July | 23.73 | 328.9 | 89.2 | This work |
| Ecc. max - P315 | 0.053 | 315 | August | 23.84 | 329.1 | 82.7 | Beaufort et al., 2022 |