# Peer review of "Eccentricity forcing on Tropical Ocean Seasonality"

_Climate of the Past, 2023_

## Author Comment (AC1)

(Reviewer comments : Our reply to the reviewer)

The authors examine the seasonality of low latitude sea surface temperature (SST) and primary productivity (PP) in a set of high-eccentricity simulations spanning various longitudes of perihelion. They find that high eccentricity leads to increased seasonality in SST as well as in PP. As a result, they find that the phasing of the SST and PP seasons alter with the timing of perihelion, leading them to introduce the concept of 'eccentriseasons' where the seasons do not stay constant with the calendar year.

This study is a timely and useful addition in the literature, and highlights the important role played by eccentricity in the tropical ocean seasonal cycle and how the phasing of the seasons relative to the calendar change as a result of precessional orbital changes. The results with SST are perhaps not as surprising given that it ties directly to the tropical insolation seasonality - Earth's axial tilt mainly provides a semiannual cycle of insolation near the equator and little in the form of an annual cycle, and so eccentricity provides the larger annual cycle forcing and especially at high eccentricity. I have no expertise with biology, but I surmise that the seasonal control of tropical PP is less well understood and so the results there is presumably novel.

Thank you for your general comment.

My main critique is that the manuscript is somewhat hard to read because there are many different threads of argument, and that there is a vagueness to some of the physical linkages being made. The writing needs to be improved. The manuscript would benefit from simplification, and that the connections be made more precisely.

In the new version, we are simplifying the text in trying to be more precise.

**Specific comments**

1. I strongly recommend the authors run a simulation with zero eccentricity if at all possible, and not use the low eccentricity simulations. Using low-eccentricity simulations as a control to compare against the high eccentricity simulations complicates things: you can't unambiguously separate out the contributions coming from orbital eccentricity and Earth's axial tilt, and moreover you have to consider the effects of different longitudes of perihelion in the low eccentricity case. A zero-eccentricity simulation solves both these problems and makes the analysis cleaner and simpler.

We agree with the reviewer's observation regarding the potential for a simpler approach, however, adopting such simplicity would compromise the realism of our study, as Earth's eccentricity never approaches zero. Instead, we utilized the lowest eccentricity observed in the past 2 million years. To validate our findings, we conducted two simulations with opposite precession to demonstrate the negligible impact of precession at such low eccentricities. Although, more complex, our approach is more realistic, and we think more elegant.

In addition, re-running additional simulations is not currently an option due to computational consideration. We would therefore stick with our initial dataset considering realistic simulations are valuable when compared to data although they complexify the analysis. We also consider than extracting the effect of eccentric alone would ideally require to re-run another batch of simulations with same orbital configuration outside of eccentricity that would be put as a very low value.

2. To make the connection between insolation and tropical SST seasonality more explicit, I recommend that the authors include a figure of TOA insolation averaged over the tropical latitudes for the high eccentricity and low eccentricity cases (or zero eccentricity case, assuming you do #1), and show the difference between them to reveal the magnitude and phasing of the annual insolation coming

from eccentricity.The link between the insolation and tropical SST should be discussed.

This is an important comment and proposition. We will include that figure in the new manuscript. We acknowledge the need for a more comprehensive discussion of the relationship between insolation and temperature in our manuscript: To address this, we will incorporate a discussion on the phase relationship between radiation and temperature.

3. The authors average over the 6 eccentricity cases in their evaluation of the seasonal amplitude for figure 3 and 4 (see lines 118-121). However, Erb et al. (2015) showed that their 'AE 'simulation (perihelion at autumnal equinox) has a significantly smaller seasonal amplitude in the Pacific cold tongue than the other cases (WS, VE, SS - see their figure 3e-h) and also compared to their zero eccentricity simulation (see figure 6c). Assuming that the IPSL model also shows similar behavior, this example demonstrates a problem with averaging over the 6 high eccentricity cases to evaluate the annual amplitudes, as it hides a lot of regional amplitude variation that can occur when the timing of perihelion is varied. Also, what if the behavior exhibited by the cold tongue occurs in other regional tropical oceans? Related to this, the amplitude of seasonality in PP (Figure 7) clearly has dependence on the longitude of perihelion in various regional oceans, making me question the wisdom of taking the average in the annual amplitude across the 6 eccentricity cases. The assumption of averaging over the 6 eccentricity cases needs more elaboration and justification if you want to keep it, but I'm wondering if the story can be made simpler and more precise if you do away with it.

The reviewer makes an important point. There are indeed some differences in the regional seasonality between the different simulations (although smaller in SST's, than in PP's). In that part of our discussion, we use the average of the 6 high eccentricity simulations to compare high and low eccentricity. We do that, not to hide any local or time difference, but to show that seasonality is larger at high eccentricity. The 6 high

eccentricity simulations, each with local highs, each with local lows, are average in order to compare the general context at high eccentricity with a low eccentricity simulation. We found no other robust way to show that. As noted by the reviewer, in Figure 7, there is a higher degree of regional variability in different precession PP simulations - it is why we shown those differences in Figure 7. To respond to the reviewer concern, we are plotting those maps also for SST's in the new manuscript similarly to PP (Figure 7).

The monthly tropical average SST (Figure 5) also indicate that the annual amplitude is high in all high eccentricity simulations. So, the local decrease seasonality of SST in the Eastern pacific when perihelion is at autumnal equinox, is, in our model, obviously compensated in other region.

We will add seasonal amplitude maps for each of the simulation separately (as a Supplement information or directly in the manuscript if we find it make our message clearer)

4. Related to point 3, part of the vagueness is that the authors analyze both regional tropical SST and tropical (area) mean SST, but the wording in the manuscript often does not clearly separate the conclusions between the two. This wording needs to be more precise. My understanding is that the conclusions (as stated in the abstract) applies to tropical (area) mean SST, and it should be stated as such. Those conclusions do not necessarily hold for specific regional SST where there can be marked differences in behavior, in particular over the Pacific cold tongue region (see point 3). This comment applies also for the analysis of PP, where there are clearly marked regional differences.

We fully agree with the reviewer: we will be more cautious in our wording in the new version of the manuscript, and we will carefully state that we are discussing the tropical mean SST and PP.

5. A claimed novelty of this manuscript is that the simulations done in this study – covering a complete revolution of precession – allows for revealing the gradual transition of the tropical ocean seasons

within the calendar year (section 4.2.2). However, both Erb et al. (2015) and Chiang et al. (2022) undertook simulations to cover a complete revolution of precession, and both noted a gradual shift of the seasons relative to the calendar with the Pacific cold tongue, and this gradual shift was a central focus of both these studies. In this aspect, this study and its revelation of gradual seasonal phase changes is not entirely novel, and credit should be given where it is due.

We are sorry the reviewer feels we do not acknowledge enough the work done by others, though we are well aware of those papers and cite them already. In our new version, we will acknowledge more often Erb et al (2015) and Chiang et al. (2022) results where appropriated.

6. Most sections appear to consist of only one paragraph, and as a consequence some paragraphs in this manuscript are quite long and cover many points (for example the Introduction).The long paragraphs make the manuscript difficult to read. I would suggest breaking them up into paragraphs, each paragraph covering one major point.

We thank the reviewer for pointing this aspect. We will re-organize the paragraphs, so the manuscript reads more easily.

We will correct all technical comments accordingly to the reviewer's suggestion.

---

## Author Comment (AC2)

(Reviewer comments : Our reply to the reviewer)

The authors present a set of climate simulations forced with several orbital configurations, including a precessional cycle and extreme cases of eccentricity. The analysis of the model output focuses on the direct influence of the eccentricity parameter on seasonal cycles of surface temperature and productivity of the tropical oceans. The authors find that a highly eccentric orbit largely amplifies annual cycles of marine surface temperature and productivity in the tropics, and that the seasons timing in these annual cycles shifts as a function of precession. To distinguish such eccentricity-enabled (or -enhanced) and precession-phased warm/cold seasons in the tropics -- different from typical axial tilt-related boreal/austral seasons -- the authors propose the name "eccentriseasons".

The study fits well within the scope of CP and is clearly motivated. It attempts to generalize previous findings for specific regions within the tropics and takes advantage of a biogeochemistry model component to discuss patterns in palaeo-productivity proxies, having key implications for understanding of low-latitude climate variability. The manuscript seems well outlined, although I also think it would be most helpful to break up some text blocks into paragraphs. I agree with another reviewer that parts of the text should make more clear some steps in the methods, as well as the spatial and temporal extent of reported statistics. Below I list some specific comments and technical details that I think should be addressed and could be helpful.

We thank the reviewer for this general comments. We will re-organize the paragraphs, so the manuscript read more clearly, and describe methods more extensively.

**Specific comments**

1.  I think it is relevant to the discussion to consider the limitations of the simulated response to orbital forcing, when pre-industrial settings of greenhouse gases and ice volume are fixed. The authors assert with confidence a seasonal increase of about 2 K in a highly

eccentric orbit, but it is important to discuss that such change could be modified (amplified or dampened) by a reorganization of atmospheric and oceanic flows in response also to the high-latitude glaciation cycles. Would such additional concurrent changes in boundary conditions change amplitude or phasing in the results? Although this is probably difficult to know without additional experiments (which I do not expect authors to run), I think it benefits the discussion to address to some extent such considerations.

The aim of this study was not to test the effect of glacial cycles on the response of precession. We however agree with the reviewer that we have to state that those have an effect on the temperature and productivity. We will address the limitation of the present simulations and discuss the glacial interglacial cycles in a new discussion paragraph.

2. The authors find surprising a lack of a marked precession signal in the mean annual primary production. I wonder if this could also be related to model constraints I think the discussion would also benefit from briefly mentioning previous palaeo-applications of the biogeochemistry model. In this case the model is being used to understand patterns in proxy data, but has palaeo-data been used to assess model performance? The comparison to modern data in the manuscript is a useful reference, but I think it would also be relevant in case there is a previous application of the biogeochemistry model to, for instance, mid-Holocene or Last Glacial Maximum conditions.

The small response of annual PP to precession was not necessarily expected (paleodata show a responsive PP), but it was not surprising since the signal is averaging. We do not think that the modest change in annual PP is due to model constraints, but more by the use of a tropical average (hint: a regional increase somewhere can be canceled out by a regional decrease somewhere else). We will add one paragraph discussing this in more detail.

This model has been used the look at productivity pattern during LGM, but we recognize that this type of framework has its own limitation because to simulate accurately the paleo-productivity you would need to be able to consider changes such as river nutrient supply or dust as

boundary conditions. This is usually not done when running orbital simulations. We therefore acknowledge some limitations in the assessment of PP and could only state that the signal we simulate is the effect of changing the oceanic circulation alone. We will introduce a limitation paragraph in the discussion.

**Technical comments**

We will correct all technical comments accordingly to the reviewer's suggestion.

---

## Author Response (AR1)

**Luc Beaufort**
Directeur de Recherche CNRS
CEREGE
1 avenue Louis Philibert
13090 Aix en Provence
Tel. +33 (0)4 13 94 91 75

[Figure]

Objet : CP-2023-80 author response

12/04/2024

Dear Erin McClymont,

Thank you for evaluating our response to the reviewers' comments on our manuscript CP-2023-80.

Please find attached the revised manuscript and the track-change file. We carefully addressed all the corrections requested by the two reviewers. Throughout the process, we made adjustments to our response in two instances:

-We agreed with reviewer 1 that it would be important to include a new figure illustrating the evolution of solar radiation during a precession cycle. Instead, we created a new Figure 2 depicting the declination of the sun and the position of the perihelion overlaid on a map of the simulated sea surface temperatures by latitudes. We believe this provides a clearer demonstration of the relationship between radiation and sea surface temperatures.

-We ultimately decided not to implement the changes proposed by Reviewer 1 regarding the original Figures 3 and 4 in their technical comments. We believe that incorporating these changes would have significantly complicated the discussion.

We have addressed all other requested changes and greatly appreciate the precision and helpfulness of the two reviews.

We hope you will find our revisions satisfactory.

Thank you for your time in editing this work.

Best regards,

Luc Beaufort and Anta Sarr